# Ex Vivo and In Vitro Antiaging and Antioxidant Extract Activity of the *Amelanchier ovalis* from Siberia

**DOI:** 10.3390/ijms232315156

**Published:** 2022-12-02

**Authors:** Lyudmila Asyakina, Victor Atuchin, Margarita Drozdova, Oksana Kozlova, Alexander Prosekov

**Affiliations:** 1Laboratory of Natural Nutraceuticals Biotesting, Research Department, Kemerovo State University, 650000 Kemerovo, Russia; 2Laboratory of Optical Materials and Structures, Institute of Semiconductor Physics, 630090 Novosibirsk, Russia; 3Research and Development Department, Kemerovo State University, 650000 Kemerovo, Russia; 4Department of Industrial Machinery Design, Novosibirsk State Technical University, 630073 Novosibirsk, Russia; 5R&D Center “Advanced Electronic Technologies”, Tomsk State University, 634034 Tomsk, Russia; 6Department of Bionanotechnology, Kemerovo State University, 650000 Kemerovo, Russia

**Keywords:** *Amelanchier ovalis*, extracts, antioxidants, geroprotectors, medicinal plants of the Siberian Federal District, phenolic acids

## Abstract

Phenolic acids are biologically active substances that prevent aging and age-related diseases, e.g., cancer, cardiovascular diseases, Alzheimer’s disease, Parkinson’s disease, etc. Cellular senescence is related to oxidative stress. The Siberian Federal District is rich in medicinal plants whose extracts contain phenolic acids. These plants can serve as raw materials for antiaging, antioxidant food supplements, and *Amelanchier ovalis* is one of them. In the present research, we tested the phytochemical profile of its extract for phenolic acids. Its geroprotective and antioxidant properties were studied both ex vivo and in vitro using *Saccharomyces cerevisiae* Y-564 as a model organism. The chromotographic analysis revealed gallic, p-hydroxybenzoic, and protocatechuic acids, as well as derivatives of chlorogenic and gallic acids. The research involved 0.25, 0.5, and 1.0 mg/mL extracts of *Amelanchier ovalis*, all of which increased the growth and lifespan of yeast cells. In addition, the extracts increased the survival rate of yeast under oxidative stress. An in vitro experiment also demonstrated the antioxidant potential of *Amelanchier ovalis* against ABTS radicals. Therefore, the *Amelanchier ovalis* berry extract proved to be an excellent source of phenolic acids and may be recommended as a raw material for use in antioxidant and geroprotective food supplements.

## 1. Introduction

Life expectancy has been increasing in recent decades. World healthcare needs novel strategies that could promote healthy aging, and diet seems to be an effective tool that is able to prevent at least some age-related diseases [1,2]. Aging is one of the most relevant issues on the global agenda. In Russia, 60-plusers made up more than a quarter of the population in 2018. By 2050, this figure will reach 31.2% [3]. As cell damage accumulates and defense stress responses weaken, cellular senescence manifests itself as mild systemic inflammation and oxidative stress. Cells stop proliferating and become dysfunctional: they release inflammatory molecules, reactive oxygen intermediates, and extracellular matrix components. Excess free radicals damage valuable biomolecules, i.e., DNA, lipids, and proteins. Such damage contributes to the development of various age-related diseases, such as cancer, Alzheimer’s disease, Parkinson’s disease, diabetes mellitus, and atherosclerosis [4,5,6].

According to the World Health Organization, the number of cancer patients is increasing; if this trend continues, the number of cancer cases in the world will increase by 60% in twenty years. This situation is exacerbated by the fact that governments have to allocate the main healthcare funds to the fight against infectious diseases, especially in low- and middle-income countries. Healthy eating habits maintain human health, working capacity, and longevity. Geroprotectors are compounds that increase life expectancy and improve its quality. Animal tests revealed more than two hundred geroprotectors [7,8]. Geriatrics are intended to treat senior citizens or improve their quality of life, while antioxidants are meant for the young and mature populations [9]. Phenolic acids are a major class of polyphenols that can inhibit excessive body damage caused by free radicals [10]. These biologically active substances have greater antiradical activity than antioxidant vitamins. Their bioavailability and water solubility are higher than those of flavonoids. Phenolic acids are absorbed in the stomach, while most flavonoids degrade under the effect of pH, digestive enzymes, and intestinal microorganisms [11].

Phenolic acids owe their antioxidant potential to an aromatic ring with a carboxyl group, -OH groups, and -OCH_3_ methoxy groups. Their antioxidant effect depends on mechanisms such as the transfer of a hydrogen atom or a single electron (proton), as well as sequential proton loss electron transfer. The latter is especially active in alkaline and biological media. The main groups of phenolic acids include hydroxy derivatives of benzoic and cinnamic acids [12]. Chlorogenic acid is a combination of caffeic and quinic acids. It is the most common soluble bound hydroxycinnamic acid. Other common hydroxycinnamic acids include ferulic, caffeic, p-coumaric, and synapic acids. Hydroxybenzoic acids share the same structure of C6-C1 and are derivatives of benzoic acid. They are soluble, i.e., conjugate with sugars or organic acids. As lignin, they are associated with cell wall fractions [13]. *n*-Hydroxybenzoic, protocatechuic, vanillic, and syringic acids are the four most common hydroxybenzoic acids. Phenolic acids are found in a variety of plant foods. Plants that grow in the Siberian Federal District are excellent sources of numerous biologically active substances, including phenolic acids [14].

*Amelanchier ovalis,* otherwise known as irga, juneberry, or shadberry, contains various health-promoting substances which can fortify functional foods and dietary supplements. These berries contain protein, pectin, insoluble and soluble fiber, vitamins, minerals, and various sugars, e.g., glucose, fructose, sorbitol, etc. Storage conditions, climate, genotype, and ripeness affect the chemical profile of these berries. Polyphenols were found in different anatomical and morphological parts of *Amelanchier ovalis*. The skin of saskatoon berries is rich in anthocyanins and phenolic acids [15]. *Saccharomyces cerevisiae* is a unicellular eukaryote that provides a good model system for aging and age-related antioxidant stress. This yeast has a relatively short and easily measurable chronological lifespan. In addition, *S. cerevisiae* has nutrient- and energy-sensitive metabolic pathways that determine longevity. The same signaling pathways were found in eukaryotes of different types [16,17]. This research featured the phytochemical profile of the *Amelanchier ovalis* extract from the Siberian Federal District, Russia, and focused on the phenolic acids with their geroprotective and antioxidant properties, which were studied ex vivo and in vitro.

## 2. Results

The research featured a dry extract obtained from a water–alcohol extract of dried Amelanchier ovalis berries. In the wild, this Amelanchier species can be found on rocky slopes, in forests, and in pine forests. The shrub grows to a height of 250 cm. Young shoots with silvery pubescence appear during the growth process. Mature branches are brown–red. The leaf blades can reach a length of 4 cm. The leaves are dark green in summer and purplish red in autumn. Inflorescences consist of large white flowers. The fruits are black with a bluish bloom. Amelanchier ovalis can be found in the territory of the Siberian Federal District in the Altai Republic, the Kemerovo Oblast—Kuzbass, the Novosibirsk and Tomsk Oblasts, as well as the Krasnoyarsk Krai (Figure 1).

### 2.1. HPLC Analysis of the Dry Amelanchier ovalis Extract

The berries of *Amelanchier ovalis* were extracted and dried, yielding a dry extract. This extract did not have a high degree of purification, but the parameters for obtaining an extract with an increased yield of the phenolic acid complex were selected in a previous study [18].

Figure 2, Figure 3 and Figure 4 and Table 1 illustrate the quantitative and qualitative content of phenolic acids in the ethanol extract of *Amelanchier ovalis*.

The qualitative and quantitative analysis of the dry extract of *Amelanchier ovalis* revealed derivatives of chlorogenic and gallic acid, as well as gallic, p-hydroxybenzoic, and protocatechuic acids. The berry extract contained more gallic acid than chlorogenic acid. The dry extract had the following acid concentrations: gallic acid—21.5 mg/g; p-hydroxybenzoic acid—20.0 mg/g; protocatechuic acid—31.0 mg/g. The gallic and chlorogenic derivatives were 20.0 mg/g and 10.5 mg/g, respectively. This result differed from previous studies where chlorogenic acid was the most abundant acid in *Amelanchier ovalis* berries [19,20]. This difference can be explained by the difference in environment, climate, storage, and extraction methods.

Since the extract is an extract of soluble substances from the plant, in addition to phenolic acids, it contains other secondary metabolites of *Amelanchier ovalis* berries. Thus, according to preliminary identification, peaks 12 and 13 represent 4-hydroxyphenyl-β-glucopyranoside (arbutin) and its derivative. The remaining distinguishable peaks presumably belong to the flavonoid group. Therefore, in further studies, it will be necessary to isolate and purify a group of phenolic acids.

Increased attention to the study of the class of phenolic acids is due to their antioxidant properties. It has been shown that acids such as chlorogenic, gallic, protocatechic, p-hydroxybenzoic, caffeic, ellagic, ferulic, and others have expanded delocalization and intramolecular hydrogen bonds. Functional groups of these substances (C=O or C=C in the groups COOH, COOR, C=COOH and C=COOR, as well as orthodiphenol functional groups) are able to stabilize some types of radicals. Among the known mechanisms of the protective activity of polyphenols in phenolic acids in water and ethanol, there is sequential proton transfer with the loss of an electron [21].

### 2.2. Effect of Amelanchier ovalis on the Growth of Saccharomyces cerevisiae Y-564

The effect of the *Amelanchier ovalis* extract on the growth of *S. cerevisiae* Y-564 yeast cells was studied at extract concentrations of 0.25, 0.5, and 1 mg/mL. Figure 5 demonstrates a 48-h growth curve that shows the growth of *S. cerevisiae* Y-564 cells. All the concentrations of the *Amelanchier ovalis* extract, i.e., 0.25, 0.5, and 1 mg/mL, had a positive effect on yeast cell growth. The maximal accumulation of yeast biomass occurred during its incubation with the extract sample of 1 mg/mL. A lower concentration of the extract (0.5 mg/mL) increased the time of yeast adaptation, but the accumulation of biomass was still higher than in the control. The sample with 0.1 mg/mL had no effect on yeast growth. All the samples did not inhibit the yeast growth and produced a positive dose-dependent effect.

### 2.3. Chronological Lifespan of Saccharomyces cerevisiae Y-564 Treated with the Amelanchier ovalis Extract

All the samples of the *Amelanchier ovalis* extract accelerated the yeast growth. The model cells demonstrated no signs of growth inhibition. The potential of the extract to increase the chronological lifespan of *Saccharomyces cerevisiae* Y-564 was analyzed according to the following scheme. The yeast samples were cultured with a 0.25 mg/mL, 0.5 mg/mL, or 1.0 mg/mL extract for 28 days. The chronological lifespan was measured by counting the number of colony-forming units (CFUs) under caloric restriction conditions. Figure 6 shows the effect of treatment time on the survival of treated and untreated yeast samples. All the concentrations increased the percentage of viable yeast cells compared to the control. The untreated control sample demonstrated a complete loss of cellular viability after 15 days of incubation. The increase in life span at 50% viability occurred in the test samples on days 1.4, 3.9, and 5.2. The increase was dose-dependent.

### 2.4. Antioxidant Properties of the Amelanchier ovalis Extract

#### 2.4.1. Hydrogen-Peroxide-Induced Oxidative Stress

Another experiment tested the effect of the *Amelanchier ovalis* berry extract on the survival rate of *S. cerevisiae* Y-564 yeast under the antioxidant stress induced by hydrogen peroxide (H_2_O_2_). Oxidative stress is one of the factors of aging resulting from an unbalanced content of radicals, in particular radicals such as H_2_O_2_. Hydrogen peroxide is capable of diffusing through the membrane of the endoplasmic reticulum, where it is formed during oxidative folding and has a negative effect on proteins. Under the action of divalent iron (Fe^2+^), a peroxyl radical can be formed from hydrogen peroxide, which is very reactive and initiates lipid peroxidation. As a result of this process, lipid and lipid peroxyl radicals are formed, interacting with polyunsaturated fatty acids. As a result, lipid peroxides are formed. Such processes lead to cell death [4]. Therefore, it is important to evaluate the effect of *Amelanchier ovalis* extract on the survival rate of living cells in conditions with high H_2_O_2_ content. Figure 7 shows the survival rate in the model organism under oxidative stress. The extract increased yeast survival when the yeast samples were treated with 0.5 and 1.0 mg/mL of the *Amelanchier ovalis* extract. During the ex vivo test, the samples of *S. cerevisiae* Y-564 treated with both the phenolic-acid-containing extract and H_2_O_2_ experienced an antioxidant effect and demonstrated greater viability than the samples treated with H_2_O_2_ alone.

#### 2.4.2. In Vitro Antioxidant Activity of the *Amelanchier ovalis* Extract

The antioxidant activity of the extract with phenolic acids was evaluated according to its ability to inhibit ABTS radicals. The *Amelanchier ovalis* berry extract showed good antioxidant properties (Figure 8, Table 2). The most effective concentration (EC_50_) of the *Amelanchier ovalis* extract was 0.42 mg/mL; this sample had the highest phenolic acid content. The concentration of ascorbic acid for 50% radical inhibition was higher by 7% (0.45 mg/mL). Therefore, the extract was able to inhibit the high antiradical activity of ABTS radicals.

## 3. Discussion

Hydrogen peroxide (H_2_O_2_) and its hydroxymethyl or hydroxyperoxide radicals induce cellular oxidative stress, alter metabolic functions, and disrupt molecular mechanisms in yeast. A lack of antioxidants that protect *S. cerevisiae* cells from stress may also affect metabolic functions and disrupt molecular events. Therefore, external antioxidants may protect stressed cells [22]. Natural antioxidants represented in this research by the phenolic acid complex in the *Amelanchier ovalis* extract can delay aging and increase lifespan by minimizing free radical toxicity in the model organism of *S. cerevisiae*. Products based on *Amelanchier ovalis*, which grows in the Siberian Federal District, are promising, as they will have a high value. This is due to the fact that the *Amelanchier ovalis* shrub is a frost-resistant and unpretentious plant. *Amelanchier ovalis* grows in nutrient-poor soils. In addition, harmful substances are practically not used. Products from *Amelanchier ovalis* are of interest for exportation to countries such as India and China located in geographic points to the Siberian Federal District.

This research is the first to demonstrate, both in vitro and ex vivo, the geroprotective and antioxidant potential of the Siberian *Amelanchier ovalis* berry extract. It featured the geroprotective properties of a dried 50% ethanolic extract of *Amelanchier ovalis* berries. This plant is also known as juneberry, shadberry, or saskatoon. Saskatoon berries are a source of beneficial nutrients and can be functional foods. For instance, Lavola et al. studied the phytochemical composition of *Amelanchier alnifolia* Nutt. grown in Finland. These berries contained cyanidin-based anthocyanins (63% phenols), quercetin-based flavonoglycosides, and hydroxycinnamic acids. Lavola et al. were the first to discover protocatechuic acid in these berries. They also found chlorogenic acid in the leaves of *Amelanchier alnifolia* Nutt. [23].

Lachowicz et al. showed that the concentration of phenol acids in the saskatoon berry genotypes ranged from 564.70 to 1216.92 mg/100 g [20]. Phenolic acids are known to have good antioxidant and antiradical properties, especially chlorogenic, gallic, and protocatechuic acids [24]. The abovementioned study by Lachowicz et al., which featured the chemical composition and properties of *Amelanchier alnifolia* Nutt. from central Poland, detected polyphenolic biologically active substances such as anthocyanins, phenolic acids, and flavonols. They identified the following phenolic acids: protocatechuic (1.87–6.10 mg/100 g), neochlorogenic (73.32–209.58 mg/100 g), p-hydroxybenzoic (5.14–42.78 mg/100 g), cryptochlorogenic (5.48–10.36 mg/100 g), gallic (6.94–16.23 mg/100 g), chlorogenic (305.85–986.14 mg/100 g), 4-caffeylquinic (20.84–97.33 mg/100 g), dicaffeic (2.02–7.87 mg/100 g), and caffeic glucoside (10.31–54.57 mg/100 g). The antioxidant activity of the berries depended on the variety and varied from 8.68 to 35.66 mmol/100 g in Trolox equivalents. Their research revealed a positive correlation between phenolic acids, flavan-3-ols, and antioxidant capacity [19]. Some recent studies demonstrated the antiradical activity of all saskatoon berry genotypes against DPPH radicals, which averaged 23.9 mmol/100 g in Trolox equivalents [15]. In our study of the *Amelanchier ovalis* berry extract, we detected the following acids: gallic (21.5 mg/g), p-hydroxybenzoic (20.0 mg/g), and protocatechuic (31.0 mg/g), as well as chlorogenic (10.5 mg/g) and gallic acid derivatives (20.0 mg/g). The difference in phytochemical composition and the predominance of specific acids in previous studies of saskatoon extracts could be attributed to the different climatic conditions of cultivation in Poland and Siberia. Despite the fact that the berries were picked in the summer, cold snaps frequently occur during this season, which can alter the chemical composition of the berries. Moreover, the phytochemical composition may vary due to different extraction methods. *Amelanchier alnifolia* was extracted by mixing the raw material with 50% ethanol and heating to speed up the infusion process. The extract had a high in vitro antioxidant capacity of 0.42 mg/mL, which is comparable to that of the standard antioxidant ascorbic acid (EC_50_ = 0.45 mg/mL).

Biologically active substances are known for a great variety of therapeutic effects with maximal activity and minimal side effects [25]. Their antioxidant properties are used against cancer, memory impairment, neurodegenerative diseases, atherosclerosis, diabetes, and cardiovascular diseases [26]. Biologically active substances are often tested on a model organism such as *Saccharomyces cerevisiae*. For example, Biradar et al. studied chebulic acid and boravinon B. They proved that phytocomponents act as antiaging molecules, which enables them to remove oxidants, increase CFU, reduce cell damage, inhibit apoptosis and necrosis, and prolong lifespan. Phytocomponents were also reported to inhibit caspase 3/7, thus increasing the lifespan of *S. cerevisiae* [22]. Curcumin was reported to increase the average and maximal lifespan of BY474 yeast [27]. Extracts of *Cimicifuga racemosa*, *Valeriana officinalis* L., *Passiflora incarnate* L., *Ginkgo biloba*, *Apium graveolens* L., and *Salix alba* are known to prolong the chronological lifespan of yeast cultures. Each of these extracts is a geroprotector that delays the onset and slows the progress of yeast aging. They are able to increase or decrease concentrations of reactive oxygen intermediates and reduce oxidative damage to cellular proteins, membrane lipids, mitochondrial, and nuclear genomes. In addition, they also improve cell resistance to oxidative and thermal stress [16]. However, Bayliak and Lushchak studied an *R. rosea* extract and proved that extracts can act as both geroprotectors and stressors that provide no adaptation to oxidative processes. The extract reduced the survival of exponentially growing *S. cerevisiae* cells under H_2_O_2_-induced oxidative stress but increased its viability and propagation rate in the stationary phase. The authors explained this phenomenon by the fact that *R. rosea* acts as a relatively strong stressor in actively proliferating cells. When exposed to H_2_O_2_, cell defense starts to fail, which leads to cell death [28]. When treated with an apple extract, *S. cerevisiae* had a longer lifespan, a lower level of reactive oxygen intermediates, and a lower sensitivity to oxidative stress. The extract also prevented the fragmentation of nuclei and mitochondria, thus protecting cells from regulated cell death [29].

Phenolic acids also possess high antioxidant potential ex vivo [30]. Chanaj-Kaczmarek et al. used *S. cerevisiae* to prove that extracts of *G. parviflora* and *F. officinalis* contained phenolic compounds and improved the viability of Δ*sod1* cells. These effects correlated with the total content of phenolic acid [31]. In our ex vivo study, the *Amelanchier ovalis* extract increased the growth of *Saccharomyces cerevisiae* Y-564, and none of the concentrations had a negative effect on the model organism. Antioxidant activity was considered in terms of reducing the negative effects of the H_2_O_2_ molecule on yeast cells by berry extract. Among all reactive oxygen species, the O^2•^-, •OH and H_2_O_2_ radicals exhibit higher pathological reactions. They are formed in various organoids such as mitochondria (ATP synthesis process), peroxisomes, and endoplasmic reticulum (oxidative folding). The •OH radical is formed in the presence of H_2_O_2_ in a two-stage process: the Haber–Weiss and Fenton reactions. The radical destroys membranes by interacting with lipids and proteins [32]. The extract concentration of 1.0 mg/mL showed the best results, increasing the chronological lifespan and survival of yeast under oxidative stress. The data obtained made it possible to slow down cellular senescence in *S. cerevisiae* Y-564 as the *Amelanchier ovalis* berry extract reduced the oxidative stress. In further studies, we plan to extract and purify the identified phenolic acids in order to define their individual biological parameters.

## 4. Materials and Methods

The experiments were carried out in the Laboratory for Biotesting of Natural Nutraceuticals of Kemerovo State University (Kemerovo, Russia). The research featured a dry extract obtained from a water–alcohol extract of dried *Amelanchier ovalis* berries purchased from Plody Sibiri LLC (Kemerovo, Russia). Berries were picked from late July to mid-August 2022. After being dried, the berries were kept in a pest-free area at a temperature of no higher than 20 °C and a relative humidity of no more than 70%.

The crushed berries were treated with 50% ethanol, with the raw material: extractant ratio being 1:10. The procedure lasted four hours at 60 °C under constant stirring in an ES-20/60 shaker-incubator. The liquid extract was dried at 90 °C in a Mini 51 Spray Dryer B-290 (St. Gallen, Buchi, Switzerland). The extract was stored in a light-protected area at temperatures ranging from 15 to 25 °C, with a moisture content of no more than 5%. The chromatographic analysis involved stock solutions of 1 mg/mL of dry extract in a 1:1 mix of isopropyl alcohol: water. The samples were filtered through filter paper with a pore size of 0.45 µm. The separation process involved a Shimadzu HPLC system with a Thermo Accucore C18 column (100 × 2.1 mm, inner diameter—2.6 μm) in a gradient chromatography mode. The mobile phase components included acetonitrile, isopropyl alcohol, and deionized water mixed with phosphoric acid to obtain pH = 3.5. The injection volume was 20 μL. The elution rate was 0.8 mL/min at a column temperature of 30 °C. The quantitative content was determined by the method of absolute graduation with Sigma-Aldrich (St. Louis, MO, USA) and Clearsynth (Mumbai, India) standards purchased from Algimed LLC (Moscow, Russia) and Trade House HIMMED LLC (Moscow, Russia).

The *Saccharomyces cerevisiae* Y-564 strain served as a model organism. It was purchased from the National Bioresource Center at the Research Center of the Kurchatov Institute (Moscow, Russia). The frozen yeast cells were thawed and grown on a complete YEPD yeast growth medium. The incubation lasted three days and was conducted in an incubator at 30 °C. After that, one yeast colony was inoculated into the liquid YEPD medium and incubated for 24 h at 30 °C. The optical density of the 24-hour culture was measured at 600 nm using a UV 1800 spectrophotometer (Shimadzu, Kyoto, Japan). The initial culture for growth analysis had an optical density of 0.1. The *Amelanchier ovalis* extract was represented by three concentrations: 0.25, 0.5, and 1 mg/mL. The dry extract was dissolved in distilled water. Its effect on the growth of yeast cells was studied by analyzing the biomass gain of the model organism. The analysis relied on a modified method, initially developed by Kavilasha and Sasidharan [17]. The yeast strain was grown on a YEPD nutrient medium at 30 °C for 24 h. The biomass gain of *S. cerevisiae* was assessed in quartz cuvettes of a UV 1800 spectrophotometer (Shimadzu, Kyoto, Japan). The yeast strain and the extracts were kept at 30 °C for 45 h under aerobic conditions with a one-hour interval. The application rate of the yeast culture into the cuvettes was 1%. The optical density was measured at a wavelength of 600 nm. The data were mean values for three independent experiments.

The chronological lifespan (CLS) analysis made it possible to determine the lifespan of non-dividing yeast cells. The analysis relied on the method described by Maruyama et al. [33]. *S. cerevisiae* Y-564 cells were cultured in a synthetic medium which included 0.17% yeast nitrogen base without amino acids or ammonium sulfate (Difco Yeast Nitrogen Base Medium). After adding ammonium sulfate (0.5%) and glucose (0.2%), the incubation lasted for 24 h. An overnight culture of *S. cerevisiae* with an initial OD600 of 0.1 was inoculated into the synthetic medium containing the *Amelanchier ovalis* berry extract in the following concentrations: 0.25, 0.5, and 1 mg/mL. The dry extract was added directly to the medium. Then, the untreated control sample and the extract-treated yeast cell culture of *S. cerevisiae* Y-564 were incubated in a shaker at 180 rpm and 30 °C. Distilled water was used to dissolve the berry extract. The cellular survival was determined by counting the colony forming units (CFUs) every three days. An aliquot of 100 mL of yeast cell suspension was placed in 900 µL of the synthetic medium and shaken for 1 min to disperse the yeast cells. A 1:10 dilution in the synthetic medium was performed in triplicate. To calculate the CFU, we placed a 100 µL aliquot of yeast cell suspension on YEPD agar, which consisted of 1% yeast extract, 2% peptone, and 2% glucose as a carbon source. The plates were incubated at 30 °C. After incubation, we recorded the number of colonies in the dilution dishes with 20–200 colonies in each. The CFU count was determined as the number of viable yeast cells in the sample. For each group, the percentage of viable yeast cells was calculated as follows: number of viable cells per 1 mL/total cells per 1 mL × 100 CFU on day. For 100% survival, there is the number of colony forming units on the third day.

The effect of hydrogen-peroxide-induced oxidative stress was studied according to the protocol developed by Tran and Green [34], with some changes. We inoculated *S. cerevisiae* yeast cells into two 50 mL Falcon-type tubes with 10 mL of YEPD medium. After inoculation, one of these tubes was treated with the berry extract at one of the following concentrations: 0.25, 0.5, or 1 mg/mL. Both tubes were incubated at 30 °C (200 rpm). The exponentially growing yeast cells, both extract-treated and untreated, were harvested in 1 mL aliquots of the same optical density after 12–14 h of incubation. They were treated with 4.0 mM H_2_O_2_ for 1 h. The yeast cells treated with 4.0 mm H_2_O_2_ underwent a qualitative viability test using point analyses. Each yeast cell suspension (5 µL) was plated onto YPD agar and stored at 28 °C. The colony counts were performed on incubation day 3. All the experiments were performed in triplicate.

The antioxidant analysis of inhibition of ABTS+ radical cations involved 7 mM and 2.45 mM solutions of ABTS and ammonium persulfate [35]. The solutions were mixed in a ratio of 1:1 and left in the dark for 16 h. The working solution was prepared by diluting the original one with distilled water to an optical density of 0.7 at a wavelength of 734 nm. Next, 3 mL of the radical working solution was added to 400 µL of the extract. Extract concentrations were 0.25, 0.50, 1.00, 1.50, and 1.75 mg/mL. After 30 min of incubation, the optical absorption of the sample was measured on a spectrophotometer at a wavelength of 734 nm. The antiradical activity was calculated according to the formula:(1)$${\mathrm{ABTS}}^{+}\;\mathrm{inhibition}\;\left(\%\right)=\left(A_{0}-A_{1}/A_{0}\right)\times 100$$
where A_0_ is the optical density of the control, and A_1_ is the optical density of the test sample.

Ascorbic acid at concentrations of 0.25, 0.50, 1.00, 1.50, and 1.75 mg/mL served as a standard antioxidant.

The results were presented as mean values of each replicate experiment ± standard deviation (±SD). The statistic comparisons relied on Student’s t-tests with *p* < 0.05 values considered statistically significant. The concentrations with a 50% inhibition (IC_50_) were calculated by interpolating them from the regression analysis. The calculations were performed using the Microsoft Excel 2007 software. The equipment belonged to the Center for Collective Use of Instrumental Methods of Analysis in the Field of Applied Biotechnology, Kemerovo State University.

## 5. Conclusions

The berry extract of *Amelanchier ovalis* proved to be an excellent component for antioxidant food supplements. It contained gallic (21.5 mg/g), p-hydroxybenzoic (20.0 mg/g), and protocatechuic (31.0 mg/g) acids. Gallic and chlorogenic derivatives were 20.0 mg/g and 10.5 mg/g, respectively. The *Amelanchier ovalis* extract demonstrated excellent antioxidant properties both in vitro and ex vivo. In fact, its antiradical activity in vitro was comparable to that of ascorbic acid. During the experiments ex vivo, the extract protected the model organism of *S. cerevisiae* from oxidative stress. The optimal concentration was 1.0 mg/mL. This concentration did not inhibit the growth of yeast cells: the final biomass was much greater than the control. In addition, the *Amelanchier ovalis* berry extract was able to increase the chronological lifespan of *S. cerevisiae* yeast.

## Figures and Tables

**Figure 1 ijms-23-15156-f001:**
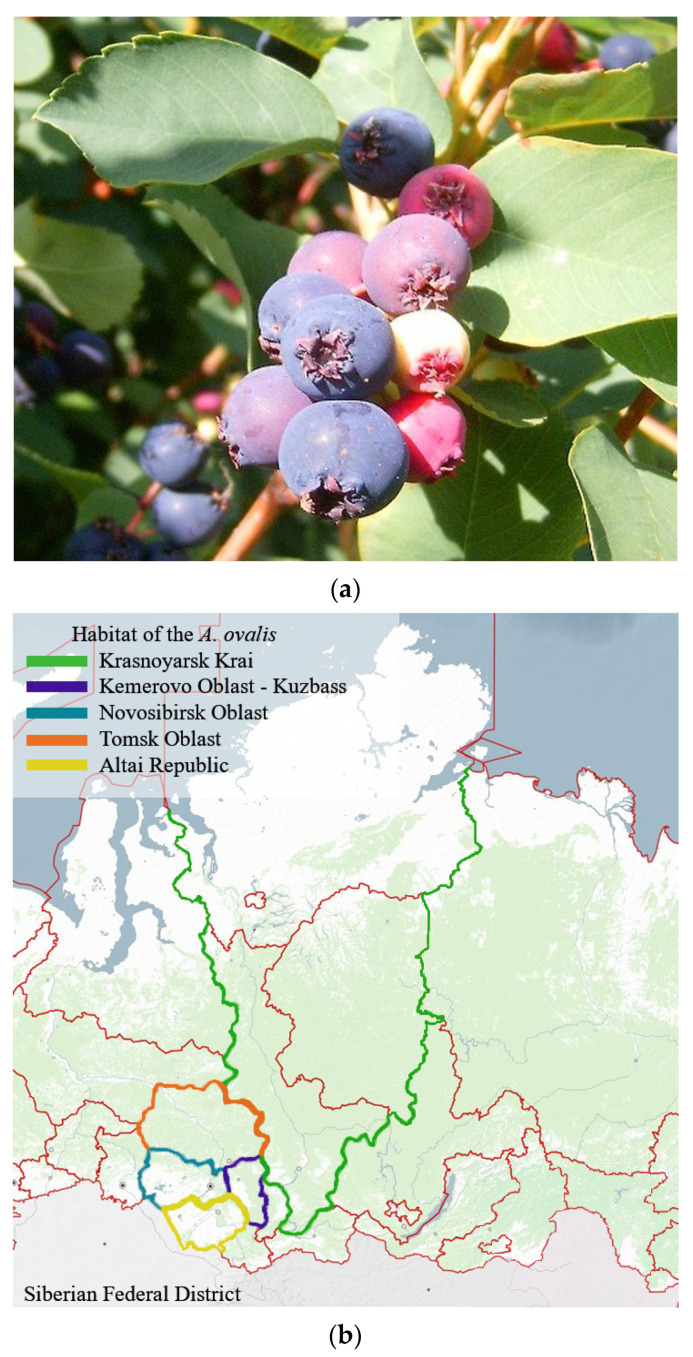
Photo of the berries *Amelanchier ovalis* (**a**) and public cadastral map of Siberian Federal District, Russia (**b**).

**Figure 2 ijms-23-15156-f002:**
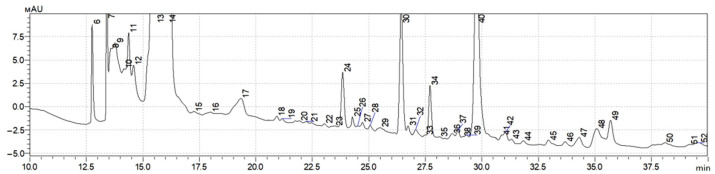
HPLC of the main polyphenols in *Amelanchier ovalis*: peak 24 is p-hydroxybenzoic acid; peak 34 is protocatechuic acid.

**Figure 3 ijms-23-15156-f003:**
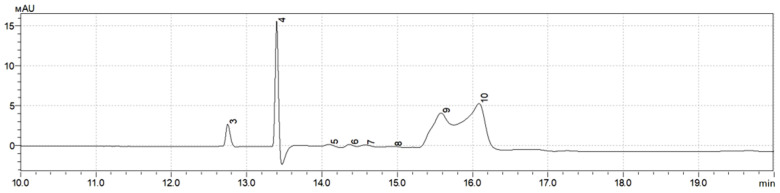
HPLC of the main polyphenols in *Amelanchier ovalis*: peaks 3 and 4 are derivatives of chlorogenic acid (320 nm).

**Figure 4 ijms-23-15156-f004:**
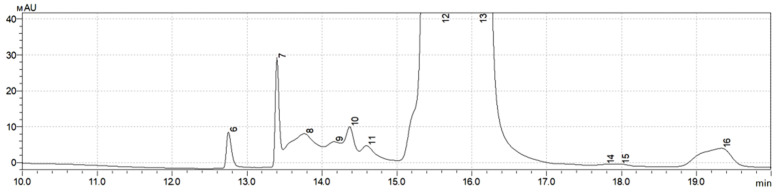
HPLC of the main polyphenols in *Amelanchier ovalis*: peak 8 is gallic acid; peaks 10 and 11 are gallic acid derivatives (277 nm).

**Figure 5 ijms-23-15156-f005:**
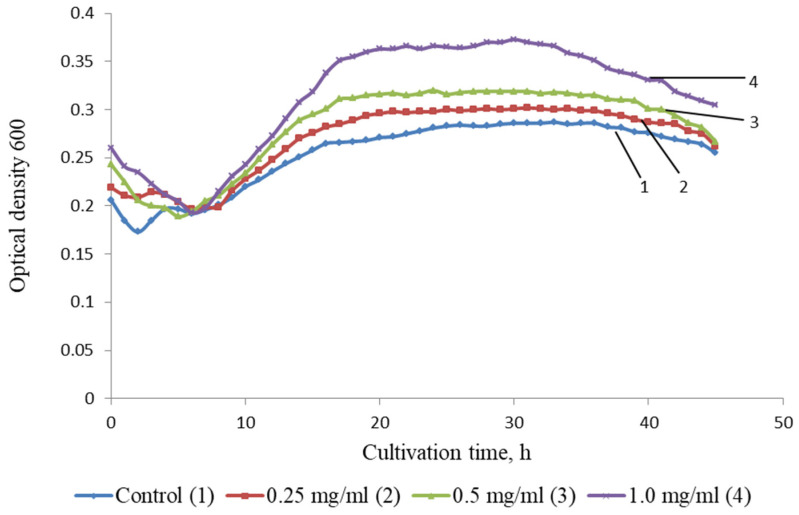
Growth curves of *Saccharomyces cerevisiae* Y-564 yeast cells with or without the *Amelanchier ovalis* berry extract: 0.25, 0.5, and 1 mg/mL.

**Figure 6 ijms-23-15156-f006:**
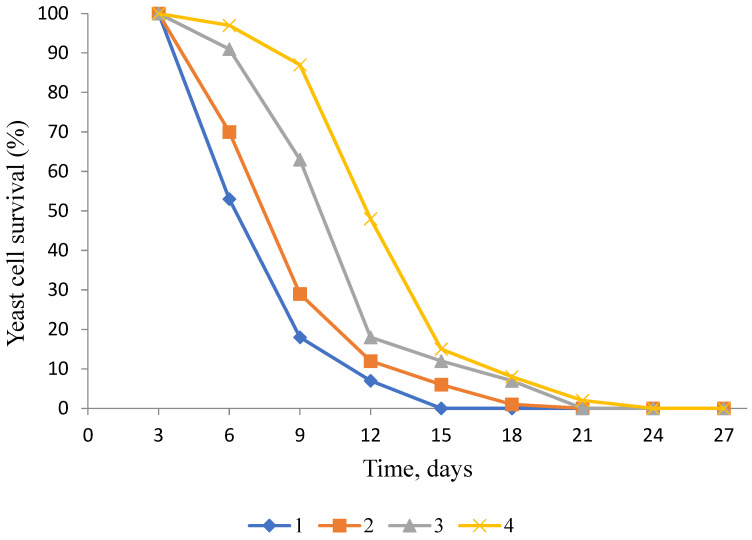
Chronological lifespan of *Saccharomyces cerevisiae* Y-564 with and without *Amelanchier ovalis* berry extract: 0.25, 0.5, and 1.0 mg/mL: 1—untreated with *Amelanchier ovalis* extract; 2—treated with *Amelanchier ovalis* (0.25 mg/mL); 3—treated with *Amelanchier ovalis* (0.5 mg/mL); 4—treated with *Amelanchier ovalis* (1.0 mg/mL).

**Figure 7 ijms-23-15156-f007:**
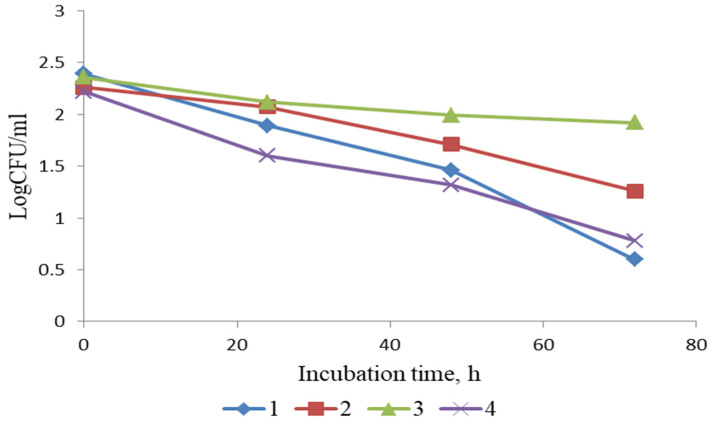
Growth curves of *Saccharomyces cerevisiae* Y-564 yeast cells with and without the *Amelanchier ovalis* berry extract: 0.25, 0.5, and 1.0 mg/mL: 1—0.25 mg/mL; 2—0.5 mg/mL; 3—1.0 mg/mL; 4—H_2_O_2_.

**Figure 8 ijms-23-15156-f008:**
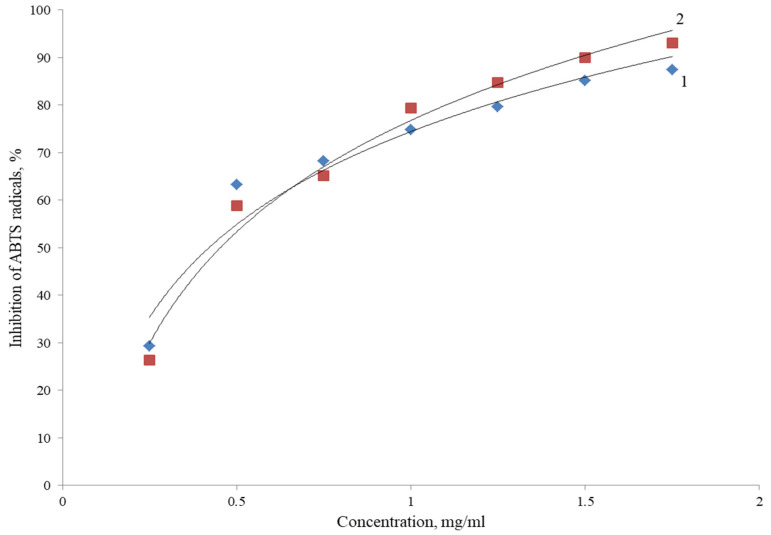
Effect of *Amelanchier ovalis* berry extract on antiradical activity: 1—*Amelanchier ovalis* extract (y_1_ = 28.189ln(x) + 74.44); 2—ascorbic acid (y_2_ = 33.805ln(x) + 76.782).

**Table 1 ijms-23-15156-t001:** Phenolic acid (mg/g) ^1^ in *Amelanchier ovalis* berry extract.

Component	Peak	Retention Time, Min	Quantity, mg/g
Chlorogenic acid derivatives ^2^	3	12,749	
Chlorogenic acid derivatives ^2^	4	13,399	10.5 ± 0.12
Gallic acid ^3^	8	13,764	21.5 ± 0.18
Gallic acid derivatives ^2^	10	14,369	20.0 ± 0.15
Gallic acid derivatives ^2^	11	14,594	
p-hydroxybenzoic acid ^3^	24	23,827	20.0 ± 0.14
Protocatechuic acid ^3^	34	27,674	31.0 ± 0.25

^1^ Mean values ± standard deviation, *n* = 3; ^2^ provisional identification by retention indices; ^3^ identification confirmed by commercial standards.

**Table 2 ijms-23-15156-t002:** Antioxidant activity of the dry *Amelanchier ovalis* berry extract.

Sample	ABTS (EC_50_ mg/mL)
Berry extract	0.42 ± 0.05
Ascorbic acid	0.45 ± 0.05

## Data Availability

Data are available from the authors on request.

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
