# Peer review of "Ex Vivo and In Vitro Antiaging and Antioxidant Extract Activity of the Amelanchier ovalis from Siberia"

_ijms, 2022, doi:10.3390/ijms232315156_

Round 1

Reviewer 1 Report

Review of "Anti-aging and antioxidant extract activity in vivo and in vitro of
the Amelanchier ovalis from the Siberia, North Asia".
This is a study investigating the chemical composition of A.Ovalis. berry, followed by analysis of its antioxidant and anti age properties. This is a nicely written paper. It's easy to read, and it provides novel information about A.Ovalis biology. This research will be of interest to specialists in the food industry. I have several comments to the authors.

Critiques/suggestions:

1)  The title is a bit misleading. The term North Asia is somewhat vague and is not commonly used. It is unclear what exactly North Asia is. I suggest excluding it from the title. Siberia is a well recognized term.

2) The choice of yeast for the in vitro experiments is justified in the introduction. However, I would recommend the authors to replicate their experiments in a mammalian cell line, which is a more representative model, considering the usefulness of these berries for human consumption.

3) Did the authors detect any HPLC peaks corresponding to env. pollutants or heavy metal-containing compounds? What is the general consensus in the field about accumulation of toxic compounds in these berries or similar berries? Does your research provide any insight on that? Please discuss this important issue in your manuscript and provide relevant data if possible.

4) It is counterintuitive to see the contents of Figure 7 at the end of the manuscript. Your Figure 7 should be your Figure 1, as it provides essential information about your study subject to the reader.

5) The title and Line 90 claims there are In vivo experiments in this study. In fact, there is none. Extract from crushed berries is ex vivo.

6) In Figure 7, please replace region with Oblast. Also, for Figure 7B, can you highlight/indicate the habitat of the berries? To avoid confusion, please clearly label Figure 7B panels in top left corner of each panel, using large font.

7) The purity of berries' extracts is unclear. What was the purity grade of your extracts?

8) In Line 76, the local name of the berry should be given (irga).

Author Response

Response 1

Review of "Anti-aging and antioxidant extract activity in vivo and in vitro of
the Amelanchier ovalis from the Siberia, North Asia".
This is a study investigating the chemical composition of A.Ovalis. berry, followed by analysis of its antioxidant and anti age properties. This is a nicely written paper. It's easy to read, and it provides novel information about A.Ovalis biology. This research will be of interest to specialists in the food industry. I have several comments to the authors.

Critiques/suggestions:

1)  The title is a bit misleading. The term North Asia is somewhat vague and is not commonly used. It is unclear what exactly North Asia is. I suggest excluding it from the title. Siberia is a well recognized term.

Thank you for the comment. We have corrected the title.

2) The choice of yeast for the in vitro experiments is justified in the introduction. However, I would recommend the authors to replicate their experiments in a mammalian cell line, which is a more representative model, considering the usefulness of these berries for human consumption.

Thank you for the recommendation. We plan to study various biological properties in the mammalian cell line at the next stages of our study. As this is a long process, we are currently unable to provide such data.

3) Did the authors detect any HPLC peaks corresponding to env. pollutants or heavy metal-containing compounds? What is the general consensus in the field about accumulation of toxic compounds in these berries or similar berries? Does your research provide any insight on that? Please discuss this important issue in your manuscript and provide relevant data if possible.

The content of heavy metals was not detected or did not exceed 0.01% according to the general pharmacopoeial monograph (State Pharmacopoeia of the Russian Federation / Ministry of Health of the Russian Federation. – XIV edit. – V.IV. – Moscow, 2018. – 1470 p.), controlling the quality of extracts. What indicates the compliance of extracts with quality according to GFM.1.2.2.2.0012.15 "Heavy metals". Together with useful phenolic acids, an admixture of the plant's secondary metabolite, 3(2-hydroxyphenyl)propionic acid, was found in trace amounts (less than 0.05 µg/kg of extract). These values correspond to the quality indicators imposed by the sanitary regulations and standards SanPiN 2.3.2.560-96 "Hygienic requirements for the quality and safety of food raw materials and food products." - intr. 2001-01-13. – M.:, 1997. to dietary supplements based on components from vegetable raw materials. No other data on the presence of toxic compounds are available. Therefore, it was decided not to consider it in further discussions.

4) It is counterintuitive to see the contents of Figure 7 at the end of the manuscript. Your Figure 7 should be your Figure 1, as it provides essential information about your study subject to the reader.

Thank you for your comment. Figure 7 has been moved to the beggining of the manuscript.

5) The title and Line 90 claims there are In vivo experiments in this study. In fact, there is none. Extract from crushed berries is ex vivo.

Thank you for the remark. We changed in vivo for ex vivo.

6) In Figure 7, please replace region with Oblast. Also, for Figure 7B, can you highlight/indicate the habitat of the berries? To avoid confusion, please clearly label Figure 7B panels in top left corner of each panel, using large font.

We have replaced region with Oblast in Figure 7. We have highlighted in different colors and indicated habitats of the berries. The panels have been moved to the upper left corner. The font has been enlarged.

7) The purity of berries' extracts is unclear. What was the purity grade of your extracts?

We have added this information to the 2.1. subsection.

8) In Line 76, the local name of the berry should be given (irga).

In line 76, saskatoon has been replaced with the local name (irga).

Reviewer 2 Report

In the present research the authors determined the main polyphenols in Amelanchier ovalis extract  from Siberia using HPLC method and studied its antioxidant and geroprotective  properties.

Comments:

Line 39-40,  you wrote: In Russia, 60-plusers made up more than a quarter of population in 2018. Please check if the statemment is correct. A quarter seems a little to me.

Materials and methods:

1.Pg 8-9,  please add:

-When the harvest was done - The voucher specimen number.

- The conservation conditions of the berries and of the extract.

You wrote at pg 2, lines 79-80 that the storage conditions affect the chemical profile of the berries.

2.Pg 10 , line 295-96 and 311: Please add in what did you solubilized the dry extract.

Results:

In Fig 3 you represented peak 9 - 3(2-hydroxyphenyl)propionic acid. Then, you didn't write anything about it. In my opinion is necessary to add a suppl material for Figures 1-3 in which to show which compounds are identified by the other peaks. For example for Fig 2 – peaks 9 and 10.

Concerning the ABTS method:

In the abstract you wrote that your research involved 0.25, 0.5 and 1.0 mg/ml extracts .

For the ABTS method you used the same concentrations? See please line 341. Why for the ascorbic acid you used  concentrations of 0-2 mg/ml?

Discussions:

1. At pg 2, lines 79-80 you wrote that  “climate affect the chemical profile of the berries” and at pg 7 lines 183-84 you wrote that “Amelanchier ovalis shrub is a frost-resistant and unpretentious plant”. Please clarify

2. Pg 8: I don't find the discussion about curcumin useful since this was not found in Amelanchier ovalis.

In my opinion it’s too much to say that Amelanchier ovalis extract has an in vivo anti-aging activity since the study was done only on yeast.

Author Response

Response 2

In the present research the authors determined the main polyphenols in Amelanchier ovalis extract  from Siberia using HPLC method and studied its antioxidant and geroprotective  properties.

Comments:

Line 39-40,  you wrote: In Russia, 60-plusers made up more than a quarter of population in 2018. Please check if the statemment is correct. A quarter seems a little to me.

Thank you for your comment. This data was found in the following source: Asyakina, L.K.; Fotina, N.Y.V.; Izgarysheva, N.V.; Slavyanskiy, A.A.; Neverova, O.A. Geroprotective potential of in vitro bioactive compounds isolated from yarrow (Achilleae millefolii L.) cell cultures. Foods and Raw Materials 2021, 9, 126–134. DOI: 10.21603/2308-4057-2021-1-126-134. In addition, statistical data from the Russian Federation's Ministry of Labor and Social Protection (Report on the results of comprehensive monitoring of the socioeconomic situation of older people in 2020. According to the data (access mode: https://mintrud.gov.ru/docs/1873), the population over the age of 60 accounted for 25.2% of the total population on January 1, 2021.

Materials and methods:

1.Pg 8-9,  please add:

-When the harvest was done - The voucher specimen number.

- The conservation conditions of the berries and of the extract.

You wrote at pg 2, lines 79-80 that the storage conditions affect the chemical profile of the berries.

We purchased shadberry berries from a private berry producer Plody Sibiri LLC (Russia). Berries were picked from late July to mid-August 2022. After drying, the berries were kept in a pest-free area at a temperature of no higher than 20 °C and a relative humidity of no more than 70%.  The extract was stored in a light-protected area at temperatures ranging from 15 to 25 °C, with a moisture content of no more than 5%. We have added this information to the Materials and Methods section (highlighted).

2.Pg 10 , line 295-96 and 311: Please add in what did you solubilized the dry extract.

We have added the information on solubilizer.

Results:

In Fig 3 you represented peak 9 - 3(2-hydroxyphenyl)propionic acid. Then, you didn't write anything about it. In my opinion is necessary to add a suppl material for Figures 1-3 in which to show which compounds are identified by the other peaks. For example for Fig 2 – peaks 9 and 10.

Thank you for the recommendation. Together with useful phenolic acids, an admixture of the plant's secondary metabolite, 3(2-hydroxyphenyl)propionic acid, was found in trace amounts (less than 0.05 µg/kg of extract). These values correspond to the quality indicators imposed by the sanitary regulations and standards SanPiN 2.3.2.560-96 "Hygienic requirements for the quality and safety of food raw materials and food products." - intr. 2001-01-13. – M.:, 1997. to dietary supplements based on components from vegetable raw materials. No other data on the presence of toxic compounds are available. Therefore, it was decided not to consider it in further discussions. Identification of substances of other peaks is a laborious process, which also requires expensive reagents, including standards. At the moment we cannot identify them.

Concerning the ABTS method:

In the abstract you wrote that your research involved 0.25, 0.5 and 1.0 mg/ml extracts .

For the ABTS method you used the same concentrations? See please line 341. Why for the ascorbic acid you used  concentrations of 0-2 mg/ml?

Concentrations of 0.25, 0.50, 1.00, 1.50, 1.75 mg/mL were used to study the antiradical activity of Amelanchier ovalis extract and ascorbic acid. In this experiment, 5 concentrations were chosen to accurately find the EC50 value, as it could have a higher concentration value. At the same time, the range of concentrations that were tested on yeast were also tested on ABTS.

Discussions:

  1. At pg 2, lines 79-80 you wrote that “climate affect the chemical profile of the berries” and at pg 7 lines 183-84 you wrote that “Amelanchier ovalis shrub is a frost-resistant and unpretentious plant”. Please clarify

We have added the clarification to the Discussions section.

  1. Pg 8: I don't find the discussion about curcumin useful since this was not found in Amelanchier ovalis.

Page 8 gives a short discussion about the prospects of using Saccharomyces cerevisiae as a model organism for testing extracts and biologically active substances for resistance to oxidative stress and increased lifespan. Therefore, we review a number of previous studies to confirm the relevance of the use of Saccharomyces cerevisiae. When re-reading the manuscript, we noticed that the main idea had been lost when the paragraph about curcumin was moved to a new line during translation. That is why we have combined this paragraph with the previous one.

In my opinion it’s too much to say that Amelanchier ovalis extract has an in vivo anti-aging activity since the study was done only on yeast.

Thank you for the comment. We have replaced in vivo withg ex vivo. Our research focuses on the preclinical development of anti-aging agents, which is why we used Saccharomyces cerevisiae in the first phase of the study. We can discuss the potential efficacy of the ex vivo anti-aging effect of the shadberry extract because Saccharomyces cerevisiae is a typical preclinical experimental model for the study of aging, as confirmed by a number of review and research articles: [Folch J, Busquets O, Ettcheto M, Sánchez-López E, Pallàs M, Beas-Zarate C, Marin M, Casadesus G, Olloquequi J, Auladell C, Camins A. Experimental Models for Aging and their Potential for Novel Drug Discovery. Curr Neuropharmacol. 2018;16(10):1466-1483. doi: 10.2174/1570159X15666170707155345. PMID: 28685671; PMCID: PMC6295931.; Longo V. D. et al. Replicative and chronological aging in Saccharomyces cerevisiae //Cell metabolism. – 2012. – Т. 16. – â„–. 1. – С. 18-31. ; Lutchman V. et al. Discovery of plant extracts that greatly delay yeast chronological aging and have different effects on longevity-defining cellular processes //Oncotarget. – 2016. – Т. 7. – â„–. 13. – С. 16542.].